# Extra-Low Dosage Graphene Oxide Cementitious Nanocomposites: A Nano- to Macroscale Approach

**DOI:** 10.3390/nano11123278

**Published:** 2021-12-02

**Authors:** Mehdi Chougan, Francesca Romana Lamastra, Eleonora Bolli, Daniela Caschera, Saulius Kaciulis, Claudia Mazzuca, Giampiero Montesperelli, Seyed Hamidreza Ghaffar, Mazen J. Al-Kheetan, Alessandra Bianco

**Affiliations:** 1Dipartimento di Ingegneria dell’Impresa “Mario Lucertini”, Università degli Studi di Roma “Tor Vergata” and Consorzio INSTM Unità di Ricerca “Roma Tor Vergata”, Via del Politecnico, 00133 Roma, Italy; Mehdi.Chougan2@brunel.ac.uk (M.C.); montesperelli@stc.uniroma2.it (G.M.); bianco@uniroma2.it (A.B.); 2Department of Civil and Environmental Engineering, Brunel University London, Uxbridge UB8 3PH, Middlesex, UK; seyed.ghaffar@brunel.ac.uk; 3Istituto per lo Studio dei Materiali Nanostrutturati, Consiglio Nazionale delle Ricerche (ISMN-CNR), Via Salaria Km 29.300, Monterotondo, 00015 Roma, Italy; eleonora.bolli@ismn.cnr.it (E.B.); daniela.caschera@cnr.it (D.C.); saulius.kaciulis@cnr.it (S.K.); 4Dipartimento di Scienze e Tecnologie Chimiche, Università degli Studi di Roma “Tor Vergata”, Via della Ricerca Scientifica, 00133 Roma, Italy; claudia.mazzuca@uniroma2.it; 5Department of Civil and Environmental Engineering, College of Engineering, Mutah University, Mutah, P.O. Box 7, Karak 61710, Jordan; mazen.al-kheetan@MUTAH.EDU.JO

**Keywords:** graphene oxide, cementitious nanocomposites, rheology, workability, mechanical properties, permeability

## Abstract

The impact of extra-low dosage (0.01% by weight of cement) Graphene Oxide (GO) on the properties of fresh and hardened nanocomposites was assessed. The use of a minimum amount of 2-D nanofiller would minimize costs and sustainability issues, therefore encouraging the market uptake of nanoengineered cement-based materials. GO was characterized by X-ray Photoelectron Spectroscopy (XPS), Fourier-transform infrared spectroscopy (FTIR), Atomic Force Microscopy (AFM), X-ray Diffraction (XRD), and Raman spectroscopy. GO consisted of stacked sheets up to 600 nm × 800 nm wide and 2 nm thick, oxygen content 31 at%. The impact of GO on the fresh admixtures was evaluated by rheology, flowability, and workability measurements. GO-modified samples were characterized by density measurements, Scanning Electron Microscopy (SEM) analysis, and compression and bending tests. Permeability was investigated using the boiling-water saturation technique, salt ponding test, and Initial Surface Absorption Test (ISAT). At 28 days, GO-nanocomposite exhibited increased density (+14%), improved compressive and flexural strength (+29% and +13%, respectively), and decreased permeability compared to the control sample. The strengthening effect dominated over the adverse effects associated with the worsening of the fresh properties; reduced permeability was mainly attributed to the refining of the pore network induced by the presence of GO.

## 1. Introduction

Worldwide, 30 billion tons of concrete are used each year, and the demand is rapidly growing. At least 8% of the global emissions come from the cement industry alone; roughly 600 kg of CO_2_ is released for every ton of cement produced [1]. Nanotechnology is included among the strategies currently proposed to reduce such a huge carbon footprint.

The development of next-generation construction materials has been in progress since the mid-2000s based on different cutting-edge approaches, including the incorporation of carbon nanotubes (CNTs) and graphene-based materials (GBMs). The former has been actively studied thus far, but investigation of the latter, mostly referring to Graphene Oxide (GO), is still at an early stage [2]. Compared to CNTs, due to the presence of oxygen-containing functional groups, GO can readily yield much more stable dispersions in water; it consists of mostly 1-nm-thick sheets, which makes it potentially more favorable for tuning mechanical, rheological, and permeability properties of the nanocomposite. Moreover, GO can be obtained in large quantities from inexpensive graphite powder. Thus, GO has attracted much attention in diverse fields, including electronics, optics, optoelectronics, bioengineering, energy storage, and purification technologies [3,4,5].

GO consists of stacked layers that contain hydroxyl, epoxide, carbonyl, and carboxyl groups. The presence of oxygen functional groups significantly alters the van der Waals interactions between the layers and endows strong hydrophilicity. According to the different models proposed by Hofmann, Ruess, Nakajima-Matsuo, and Lerf-Klinowski [6], GO contains oxygen functionalities on the basal planes and edges of sheets that, upon converting carbons from sp^2^ to sp^3^, destroy the p–p electronic conjugation and result in a significant decrease in electrical conductivity. Moreover, since GO is hydrophilic, interlamellar water molecules are always present in interlayer voids even after prolonged drying. Computational studies and experimental measurements assessed that the typical interlayer distance in GO ranges between 6.0 and 11.0 Å, depending on the relative humidity [7]. For GO monolayers, theoretical studies assessed that Young’s modulus (E) ranged between 325 and 670 GPa and intrinsic strength from 31 to 63 GPa [8]. Suk et al., using Atomic Force Microscopy (AFM) measurements combined with finite element analysis, evaluated the E value of GO monolayers around 200 GPa [9].

The key to understanding GO paper’s mechanical properties lies in examining the collective behavior of stacked platelets and the interlayer adhesive (water molecules), the properties being essentially controlled by hydrogen-bond networks through the oxygen functional groups and/or the interlayer water molecules. Since moisture content controls both the extent and the collective strength of interlayer hydrogen-bond networks, which is, in turn, manifested in the overall macroscopic response of these materials, dry GO sheets have to be considered stiffer than moist ones. Furthermore, stiffness increases with the “density” of oxygen functional groups, promoting superior interlayer adhesion. Based on these findings, interestingly, Nikhil V. Medhekar et al. in 2010 suggested the possibility of tuning the properties of GO nanocomposites by altering the density of functional groups on individual platelets, the water content, and possibly the functional groups participating in hydrogen bonding with interlayer water molecules [10]. Presently, the toxicological effects of GO are still undefined. Differences in experimental designs and/or in the physical/chemical properties of the chosen investigated product have inevitably led to dissimilar and even contrasting results [11]. In this respect, some researchers have recently been focusing on “oxidation debris” (OD), small carbonaceous fragments that originate from the high-temperature oxidation process of carbon nanostructures driven by oxidant acids (i.e., H_2_SO_4_, HNO_3_). OD remain strongly adsorbed onto the surface of the oxidized carbon nanostructures through p-p stacking interactions between aromatic rings or van der Waals interactions, promoting electrostatic stabilization in water and enabling anchoring of diverse functional groups. Some authors have proven that OD can be removed by treating GO suspensions with NaOH aqueous solution [12,13,14,15]. It is worth mentioning that this topic has been the object of passionate discussion in the field of GO chemistry in the last few years [16,17].

Over the past decade, several authors have demonstrated that incorporating high-surface-area GO nanosheets in a cement matrix effectively increases the mechanical properties of cement pastes, mortars, concrete, and even ultra-high-strength concrete in opposing the formation of defects at the nanoscopic level [18,19]. Furthermore, Mohammed et al. showed a successful implementation of GO toward the resistance of freeze-thaw cycles in cementitious materials [20]. Moreover, it is well known that high-strength concrete is more susceptible to damage at high temperatures than ordinary concrete and thus likely to undergo “spalling” due to the difficulty in releasing the accumulated water vapor trapped in its typically dense matrix. In order to create an enhanced engineered pore network able to cope with high-temperature effects, many researchers focused on the pore-refinement effect driven by the incorporation of GO in cement-based materials [21].

It can be concluded that, despite numerous encouraging outcomes, further research efforts are clearly still needed in this area to reach a cost-effective, environmentally friendly, and safe translation of GO-modified cementitious materials to the market. However, according to the most up-to-date literature, cement-based nanocomposites loaded with an extra-low dosage of GO have been scarcely investigated [2,22]. Aimed to fill this gap of knowledge, the Authors herein present a research paper focused on the impact of 0.01% (by weight of cement) of commercially available GO nano-powder on the properties of fresh admixtures and the resulting hardened cement-based nanocomposites. In view of experimental reproducibility and macro- to nano-scale understanding, the selection of the nanofiller was based on a complete structural and chemical investigation.

## 2. Materials and Methods

### 2.1. Raw Materials

A premixed commercial M5-mortar, which meets EN 998-2, containing two parts of graded sand as aggregate and binders comprising of one part Portland cement type (I) and 0.75 parts of hydrated lime, was used. The presence of hydrated lime improves the workability of mortars, making the product easier to handle for unprofessional users [23]. According to the supplier, the maximum particle size of this product is below 3 mm.

GO (Sigma Aldrich) in the form of free-flowing dry powder was employed as a nanofiller.

With the aim of developing a reproducible protocol, a ready-to-use and easy to handle commercial mortar was chosen.

### 2.2. Characterization of GO

The nanofiller was fully characterized utilizing the following techniques:

Thermal analysis was performed on ~5 mg of sample in nitrogen flow (40 mL min^−1^), in the temperature range 25–1000 °C with heating rate 5 °C min^−1^ (TGA/DSC 1, Mettler Toledo).

Fourier-transform infrared spectroscopy (FTIR) of GO was conducted on a Thermo Fisher-Scientific Inc. (Madison, WI, USA) instrument (model iS50), equipped with a single reflection attenuated total reflectance (ATR) diamond cell. Spectra were collected from the powder without any pre-treatment; therefore, each sample was put directly on the ATR crystal. After that, 32 scans at a resolution of 4 cm^−1^ in the 4000–500 cm^−1^ region were collected for each sample. All experiments were performed in triplicate.

X-ray diffraction (XRD) patterns of GO powder were recorded in the range of 2*θ* = 5–70°, with a step size of 0.020°, time per step = 2 s, scan speed = 0.01°/s, using a Philips X’Pert Pro diffractometer with Cu Ka radiation (*λ* = 1.5405600 Å). The average height of GO stacking nanostructure (H) was evaluated from the full width at half-maximum (FWHM) of the diffraction peak at 2*θ* 11.2836°, corresponding to the {001} family of lattice planes of GO, employing Scherrer’s Equation (1) [24,25]:(1)$$H=\frac{Κ\lambda }{\beta \cos\theta }$$
where *K* is the shape factor equal to 0.9, *λ* is the wavelength of the Cu *Kα* radiation, *β* is FWHM (in radians), which is corrected for the instrumental broadening and *θ* is the Bragg angle.

Then, the number of stacked layers (N) of GO was evaluated by Equation (2):N = H/d(001)(2)
where d(001) is the interlayer distance of the {001} family of lattice planes.

X-ray Photoelectron Spectroscopy (XPS): The samples for XPS analysis were prepared by pressing the GO powder onto grated Au foil (purity 99.99%). Photoemission spectra were acquired using an Escalab 250Xi spectrometer (Thermo Fisher Scientific Ltd., East Grinstead, UK) with a monochromatic Al *Kα* (1486.6 eV) excitation source and six-channeltron detection system. All the spectra were collected at a pass energy of 40 eV in the standard mode of the electromagnetic lens system corresponding to an analysis area of approximately 1 mm in diameter. The binding energy (BE) scale was corrected for a low sample charging by positioning the main C 1s peak of C-C bonds at BE = 284.6 eV and controlling if the Fermi level corresponded to BE = 0 eV. X-ray-induced Auger spectra of the C KVV region were acquired at a pass energy of 100 eV to increase the signal-to-noise ratio. After the acquisition, the spectra of C KVV were smoothed at least 11 times by moving the average routine with a width of 1.2 eV. Afterward, these spectra were differentiated by using a width of seven data points to determine the D parameter [26]. Spectroscopic data were processed by Avantage v.5 software (Thermo Fisher Scientific Ltd., Madison, WI, USA).

Raman measurements were carried out using a Renishaw RM 2000 (Gloucestershire, UK), with a 785 nm laser source. For the analysis, GO dispersion in water was prepared with a concentration of about 10 μg/mL, sonicated for 2 h in an ultrasound bath cleaner at 80 kHz (Elmasonic P60H, Singen, Germany). The GO dispersion was analyzed in dynamic mode, with 10 accumulation points per second and a resolution of 2 cm^−1^, objective 50× and 100% of laser power. The spectrum was processed by Fityk 0.9.8 software, subtracting a linear background normalization.

AFM: A few drops of the previously described GO dispersion were deposited onto mica and characterized by a Dimension 3100 equipped with a NanoScope IIIa controller (Veeco, Santa Barbara, CA, USA) in tapping mode.

### 2.3. Fresh Admixtures and Hardened Nanocomposites: Experimental Procedure

Following the supplier’s suggestion, the control mixture was produced by adding the required amount of water (18% by weight of premixed mortar) into the dry premixed mortar then stirring at 500 rpm for 5 min using a Eurostar Digital IKA^®^-Werke (Germany) equipped with a spiral stirrer (IKA^®^-Werke, Staufen im Breisgau, Germany). For the nanocomposite admixture modified with 0.01% by weight of cement of GO, GO powder was previously dispersed into the requested amount of mixing water for 30 min using ultrasonic processor (VC750, Sonics and Materials, Newtown, CT, USA) and then added to the dry premixed mortar. The mixing protocol was the same for the control mixture (stirring at 500 rpm for 5 min).

The obtained fresh mixtures were then cast in 40 mm × 40 mm × 160 mm metallic molds, mechanically vibrated for 3 min (Retsch, Haan, Germany), and kept at room temperature under a wet towel for 24 h in accordance with ASTM C348-02. Finally, the hardened samples were demolded and kept for 7, 14, and 28 days in water at ambient temperature. For the sake of comparison, aiming to assess the impact of GO on the hydration kinetics, only the control sample was also hardened in water for 200 days.

### 2.4. Characterization of the Fresh Admixtures: Rheology, Flowability, and Workability

Rheology measurements were performed immediately after mixing using a Kinexus Lab + rheometer (Malvern Instruments Ltd., Malvern, UK) equipped with rSpace software (Malvern Panalytical Ltd., Malvern, UK). The rheology measurements were carried out in the shear rate range of 0.1 s^−1^–100 s^−1^ over ten intervals, and the apparent viscosity and shear stress (*τ*) *versus* shear rate (*γ*) curves were recorded. Among the proposed fitting models, owing to the pseudoplastic behavior of the cement-based mortars, the modified-Bingham model (Equation (3)) was chosen to determine the rheological parameters, i.e., the plastic viscosity (*η_p_*) and the yield shear stress (*τ*_0_). The chosen model showed an adequate fitting accuracy of R^2^ > 0.9987.
(3)$$\tau =\tau_{0}+\eta_{P}\cdot \gamma +C\cdot \gamma^{2}$$
where: *τ*, *τ*_0_, *η_p_*, *γ* and *C* are the shear stress, yield shear stress, plastic viscosity, shear rate, and a regression constant, respectively.

Flowability of mixtures over time was evaluated by flow table tests carried out after 0, 5, and 15 min after mixing in accordance with BS EN 1015–3:1999. For the sake of comparison, Equation (4) was used to calculate the flowability percentage.
(4)$$F\left(\%\right)=\left(\frac{D_{average-\;}D_{0}}{D_{0}}\right)\times 100$$
where *F* (%) is flow percentage, *D*_0_ represents the bottom cone diameter, and *D_average_* is average spread-diameter of the paste spread in two perpendicular directions.

Workability of cementitious composites was assessed under BS EN 12350–2:2009 standard using a mini-slump test (cone geometry: 19 mm top diameter, 38 mm bottom diameter, and 57 mm height).

In accordance with ASTM C1437, the relative slump value was calculated using Equation (5).
(5)$$\Gamma_{P}={\left(\frac{D_{average}}{D_{0}}\right)}^{2}-1$$
where, *Γ_P_* represents relative slump value, *D*_0_ bottom cone, and *D_average_* is average spread diameter of the paste spread in two perpendicular directions.

Setting time: Following ASTM C191-08 and BS EN480-2:2006, the Vicat needle test was employed to determine cementitious mixes’ initial and final setting time.

### 2.5. Hardened Nanocomposites: Density, Mechanical Tests, and Microstructure

Density was estimated by weight and size measurements by means of an analytical balance (Mettler-Toledo Ltd.) and a digital caliper.

In accordance with BS EN 196–1:2016, three-point bending tests (MTS, Eden Prairie, MN, USA) and compression tests (Matest, Treviolo, Bergamo, Italy) were performed on samples cured at 7, 14, 28, and 200 days.

Scanning Electron Microscopy (SEM; FE-SEM, LEO Supra 35, Oberkochen, Germany) was conducted on portions of samples broken in compression tests into an approximate size 5 mm × 5 mm × 5 mm. In order to stop the hydration, all samples were oven-dried for 72 h at 110 °C, stored in ethanol, and dried in air at room temperature for 12 h before SEM analysis. Specimens were previously gold coated by sputtering (EMITECHK550X sputter coater, Quorum Technologies Ltd., Lewes, UK).

### 2.6. Hardened Nanocomposites: Initial Surface Absorption Test (ISAT), Volume of Permeable Voids (VPV), and Chloride Ion Penetration

*ISAT* was performed according to BS 1881-208:1996. A batch of three cubic specimens (100 mm × 100 mm × 100 mm) cured for 28 days was tested. Previously, samples were dried at 105 ± 5 °C for 24 h, and the top surface was sealed by a 200 cm^2^ circular cap. Then, penetration of deionized water through the sample’s top surface was allowed for 10, 30, and 60 min. The ISAT rate was measured using Equation (6) [27]:(6)$$f=\frac{60}{t}\times D\times 0.01$$
where f is the initial surface absorption rate (mL/m^2^/s), D is the number of scale divisions during the test, and t is testing time (s).

Boiling-water saturation method was conducted on a batch of three specimens of size 40 mm × 40 mm × 160 mm cured at 28 days, in accordance with ASTM C 642, intending to evaluate permeable porosity. Each specimen was weighed after (i) 48 h of oven-drying at 110 °C; (ii) 48 h of submerging in tap water followed by 5 h of immersing in boiling water to determine, respectively, the oven-dry mass, saturated mass after immersion, and boiling. Based on the aforementioned data, the VPV was calculated using Equation (7) [28,29]:(7)$$\mathrm{VPV}\;\left(\%\right)=\left[\left(B-A\right)/\left(B-C\right)\right]\times 100$$
where A (g) is the mass of the oven-dried sample, B (g) is the mass of the surface-dry sample after immersion and boiling, and C (g) is the apparent mass of the saturated sample in water after immersion and boiling.

Chloride ion penetration was assessed using a salt ponding test in accordance with BS 14629:2007 with some modifications. A batch of three 100 mm × 100 mm × 100 mm cubic specimens cured for 28 days in water was previously oven-dried at 110 °C for 24 h and then exposed to a 5 wt.% sodium chloride water solution. After 30 days of exposure followed by 24 h of drying at 110 °C, samples were drilled at depths of 5, 10, 15, 20, and 25 mm, and resulting powders were collected. Silver nitrate water solution (0.02 M) was employed as the titration agent, and the chloride ion content (CC) was determined with Equation (8) [30,31]: (8)$$\mathrm{CC}\;\left(\%\right)=3.545\times F\times \left(V2-V1\right)/M$$
where F is the molarity of the silver nitrate solution, V2 is the ammonium thiocyanate solution volume used in the blank titration (ml), V1 is the ammonium thiocyanate solution volume used in the titration (ml), and M is the sample mass (g).

## 3. Results and Discussion

### 3.1. Characterization of GO: Chemical Composition, Morphological and Structural Properties

AFM analysis of highly diluted GO dispersion (see Section 2.2) enabled the accurate evaluation of GO’s lateral size and thickness.

The AFM images show well-dispersed GO sheets with maximum lateral dimensions of about 600 nm × 800 nm. Some wrinkled and overlapping GO sheets were also observed (Figure 1a,b). Single and multiple GO sheets are clearly discernible from the optical contrast in the AFM images, where the thicker multilayer structures give a whitish color from their reflection originating from the lack of transparencies. Figure 1c shows a GO flake in detail, its height analysis indicates a thickness of about 2 nm (Figure 1d) that is consistent with a stacking nanostructure made of two GO sheets [32]. 

The chemical composition of GO powder was assessed by XPS, the results of which revealed the presence of four species of carbon, as shown in Figure 2. The main component A of C 1s spectrum at BE = 284.6 eV, corresponds to C-C bonds in GO, graphene or graphite [26]. The second component B at BE = 286.4 eV is typical for C-O bonds in GO associated with epoxide and hydroxyl groups and is usually accompanied by a lower component C of the carbonyl groups at BE = 287.9 eV [33,34]. The low-intensity peak of the last component D is attributed to low content of carboxylates [32]. The total carbon content in GO amounts to 67.9 at%, as reported in the XPS quantification Table 1. Low content of sulphate is probably a residue derived from the Hummer’s process [35].

Figure 3 presents the Auger spectrum of the C KVV region in the form of the first derivative. Two peaks corresponding to two values of the D parameter [26]: D = 13.5 and 19.0 eV. The first diamond-like value is characteristic for graphene [26], whereas the second one is lower than the D parameter in graphite and could be attributed to GO [33,34]. Therefore, the electronic configuration of carbon in this sample indicates the mixture of graphene and GO.

The presence of oxygen-containing functional groups in GO was further assessed by FTIR analysis (Figure 4).

The strong broad band in the range 2700–3700 cm^−1^ is due to O–H stretching in alcohols, carboxylic acids, and water [36]. The peak at 1725 cm^−1^ is associated with C=O stretching of both carbonyl and carboxylic groups, whereas the sharp peak at 1582 cm^−1^ is characteristic of C=C skeletal vibrations of non-oxidized domains and/or of asymmetrical stretching of C=O in carboxylate salts [37,38,39]. The peak at 1399 cm^−1^ may be attributed to the O-H bending of tertiary alcohols as well as symmetric stretching of C=O in carboxylates salts [36,40]. Absorptions in the region of 900–1300 cm^−1^ may arise from C-O vibrations of several species (i.e., ethers, carboxylic acids, alcohols, epoxides, and ketones) [38,41]. The peak at 580 cm^−1^ has to be assigned to vibrations of S–O bonds of sulphate, the residue of Hummer’s reaction for the production of GO [35], also detected by XPS analysis (Table 1). 

The thermal stability of GO powder was investigated by thermal analysis (Figure 5). Four steps of weight loss at increasing temperatures were recorded: weight loss (11 wt.%) up to 120 °C associated with the removal of water molecules (adsorbed on the hydrophilic GO surface and intercalated between GO sheets); two weight losses between 120 °C and 310 °C (32 wt.%), attributed to the decomposition of labile functional groups (hydroxyl and carboxyl); and, finally, weight loss (15 wt.%) occurring at T > 310 °C ascribed to the decomposition of more stable functional groups (epoxide and carbonyl) [42,43,44,45]. 

Structural characterization of GO was performed by XRD and RAMAN. The XRD pattern of the powder shows the characteristic GO peaks at 2θ 11.28° and 42.58° due to (001) and (100) reflections, respectively (Figure 6) [46,47,48,49].

The XRD analysis showed an average height (H) of GO stacking nanostructure of 6 nm, consisting of 7–8 layers with a d-spacing of 0.7835 nm. The evaluated interlayer distance is in accordance with literature reporting values ranging between 0.6 nm and 1.1 nm [7,25]. The discrepancy between the thickness of the GO stacking nanostructure (6 nm) and that one evaluated from AFM analysis (2 nm) has to be ascribed to the different sampling employed for the two techniques as detailed reported in the 2.2 paragraph. 

The Raman spectrum of GO in the spectral range of 800 cm^−1^ to 2000 cm^−1^ shows two bands: the G band at 1591 cm^−1^, slightly blue-shifted with respect to monolayer graphene due to the presence of oxygen functionalities, in accordance with literature data [50,51,52]; and the D band located at 1313 cm^−1^ (Figure 7). The presence of the D band clearly also indicates the presence of some defects and disorders in the sp^2^-hybridized carbon-atoms lattice, which “light” it up due to the symmetry breaking and the corresponding change in the selection rules. 

Another small band is visible at about 1131 cm^−1^, referred to as D*, that is related to disordered graphitic lattices associated with the presence of some amounts of carbon atoms with sp^3^ hybridization (oxidized domains) [53]. The position of the maximum of the D* band lies in the range between 1112 cm^−1^ and 1175 cm^−1^, depending on the oxygen content, shifting to a shorter wavelength when the oxygen content increases [54]. Comparing our data with literature, the D* band position at 1131 cm^−1^ is consistent with an oxygen content of 31%, as determined from XPS analysis [54].

Generally, for graphite and graphene-based materials, broad bands are also visible in the range 2000–3000 cm^−1^, attributed to the 2D bands related to the stacking order of the layers along the c-axis [50]. The absence of clear visible peaks in that range is due to the breaking of the stacking order caused by the presence of intercalated water molecules [55].

### 3.2. Fresh GO-Modified Cementitious Admixtures: Rheology, Flowability, and Workability

The effect of the incorporation of GO in cementitious fresh admixtures has been extensively investigated because of the expected superior dispersibility attributed to the presence of oxygen functional groups [56,57]. 

Rheological flow curves obtained for the control and GO-0.01 fresh mixtures are reported in Figure 8a,b. Shear stress rises at an increasing shear rate (Figure 8a) for both samples; the curve of the GO-modified mixture results shifted upwards concerning the control, in agreement with the literature [58]. Regarding apparent viscosity (Figure 8b), both fresh mortars showed the typical downward trend from low to high shear rate [58,59]. The apparent viscosity of the GO-modified fresh mixture is higher with respect to the control mixture within the overall investigated shear rate range. Rheological characterization showed that low dosage of GO (0.01 wt.% by weight of cement) increased both the yield shear stress (*τ*_0_) and the plastic viscosity (*η*_p_) of the fresh mortar, the values being 22.46 Pa and 0.302 Pa·s and 16.60 Pa and 0.174 Pa·s for GO-0.01 and the control sample, respectively. 

The flowability registered for the control at 0, 10, and 15 min was 66.5%, 68.5%, and 69%, respectively. The incorporation of 0.01% of GO by weight of cement in the fresh mixture induced the reduction of flowability, the values being 41.5%, 43%, and 43.5% at 0, 10, and 15 min, respectively (Figure 9). Workability was evaluated through mini-slump test, and results clearly show that the slump value (Γp) registered for GO-modified admixtures decreased by one-fifth with respect to the control sample (Figure 9). 

These overall findings (i.e., the loss of flowability and workability) have to be associated with (i) the wetting of GO particles characterized by high specific surface area and strong water-absorbing capacity, (ii) the formation of aggregations of GO nanosheets by chemical cross-linking with calcium cations able to entrap large amounts of water, that lead to the shortage of free water within the mixture and thus to increased friction among the particles [57,60].

Figure 10 depicts the Vicat penetration test results of control samples and GO-0.01 over time. The initial and final setting times are represented by the intersection of curves with the two straight lines at 36 mm and 2.5 mm, respectively. The control sample’s initial and final setting times are recorded at 142 min and 224 min, respectively. At the same time, the initial and final setting time of GO-0.01 reduced to 124 min and 185 min, respectively. As can be seen, the setting time values of the mixture modified with graphene oxide (i.e., GO-0.01) are relatively lower than the values registered for the control sample. The reason has to be associated with the ability of GO particles to accelerate the hydration process of cementitious composites. 

Such drawbacks were fully compensated by the strengthening effect induced by GO incorporation into the cementitious matrix (see the following Section 3.3.1). 

### 3.3. Hardened GO Nanocomposites

#### 3.3.1. Density, Mechanical Properties, and Microstructure

In Figure 11a–c, the density and mechanical properties (compressive and flexural strength) at 7, 14, and 28 days of mortar modified with 0.01% GO are presented and compared with the respective control samples. For the sake of hydration kinetics evaluation, the results of the control sample hardened for 200 days are also reported. All the samples modified with GO experienced a density enhancement compared to the respective controls. Density progressively increased from 7 to 28 days, reaching at 14 days values comparable to those obtained for the control at 200 days. The control sample showed a peculiar densification behavior, i.e., density decreases from 14 to 28 days and then increases at 200 days. Such a trend has to be explained considering that cementitious samples’ mass and volume decrease over the hydration time. At 14 to 28 days, the effect due to the mass reduction dominated over the volume decrease of the samples; at 200 days, the reverse phenomenon occurred, and density increased [29,61]. 

In Figure 11b the results of compressive strength (Rc) are reported. A gradually increasing trend from 4.51 MPa at 7 days to 6.1 MPa at 28 days was observed.

An analogous trend was recorded for flexural strength (see Figure 11c) that reached a maximum value of 1.75 MPa at 28 days, slightly higher than the 1.66 MPa measured for the control sample at 200 days.

Such noticeable enhancement of the mechanical properties (29% and 13% increase in compressive strength and flexural strength, respectively, compared to the control sample) of GO-modified cementitious nanocomposites have been previously assessed by several studies [62,63,64]. However, it is worth mentioning that most of the literature regarding these GO-modified nanocomposites focuses on a higher GO dosage that generally ranges between 0.02 and 1% by weight of cement [2,62,65,66]. In this respect, interestingly, the observed Rc trend is in complete agreement with the data reported in the review by Zhao et al. [2]: most of the mortars containing GO in the range 0.01–0.2% by weight of cement had, at 28 days, increased compressive strength between 5 and 78% (average about 25%), the cited lower and upper values obtained for 0.01% GO and 0.1% GO, respectively.

According to the same review [2], few data are available in the literature regarding the bending strength (Rb) of mortars modified with GO whose dosage mostly ranged between 0.04 and 0.1% by weight of cement. It is reported that Rb improved from 10% to 80% for GO dosages of 0.04% and 0.03%, respectively. Therefore, the herein reported results on Rb have to be considered thoroughly in line with the literature. 

The main reinforcing mechanisms of GO on cement-based materials as proposed by the literature can be summarized as follows: (1) GO has excellent mechanical properties; (2) the seeding and pore-filling effects lead to refinement of pore structure; (3) the chemical bonding at the GO-cementitious matrix interface improves load-transfer efficiency [2,67]. The results herein obtained suggest the achievement of a uniform dispersion of the nanofiller and strong interfacial adhesion between GO and the cementitious matrix.

Figure 12 shows SEM images of the GO-cementitious nanocomposite (GO-0.01), showing the typical honeycomb morphology of C-S-H, portlandite polygonal crystals, ettringite needles, and capillary pores with size less than 2–2.5 μm. 

#### 3.3.2. Permeability Properties

The ISAT rates of the sample modified with GO ranged between 0.28 and 0.26 mL/m^2^/s, remarkably lower than those obtained for the control specimen, which varied between 0.62 and 0.46 mL/m^2^/s (Figure 13a).

In agreement with ISAT, a lower VPV was recorded for the GO-0.01 sample compared to the control (Figure 13b). 

Several studies have reported the effectiveness of GO incorporation in cementitious composites in acting as a protective cloak and increasing the resistance to the penetration of fluid into the cement-based materials [68]. It has also been hypothesized that GO sheets and their aggregates can fill the capillary pores of the cement matrix, preventing the ingress of aggressive chemicals within the structure [68]. 

A salt ponding test was carried out to assess the chloride ion penetration of the GO-modified mortar cured for 28 days (Figure 14). Following the previously reported trend of the water absorption tests, the results revealed a significant decrease of chloride ion penetration in the cementitious nanocomposite compared to the control.

These overall results suggest that the presence of an extra-low dosage of the chosen GO is expected to induce increased durability in cement-based manufacts. 

Several experimental studies have been conducted on the transport properties of GO-modified cementitious composites exposed to aggressive environments. Mohammed et al. mentioned that GO influences the transport properties along with the pore size and distribution in the cement matrix, which effectively hinders the ingress of water and chloride ions. They reported that the maximum chloride diffusion reduction of about 80% was achieved by adding 0.01% GO (by weight of cement), with the penetration depth diminishing from 26 mm to 5 mm [69]. Sharma et al. also reported the impact of GO in terms of reducing total porosity and average pore size, which are attributed to its ability to fill the micropores, leading to a microstructure characterized by enhanced packing density and fine porosity [70]. Li et al. reported the effectiveness of GO aggregates in creating more tortuous paths for the ingress of water with respect to GO nanosheets due to the higher aspect ratio [67]. 

## 4. Conclusions

The objective of this study was to evaluate the effectiveness of extra-low dosage GO cementitious nanocomposites. The chosen formulation (i.e., 0.01% by weight of cement), rarely investigated in the literature, could promote the uptake to the market of graphene-engineered construction materials owing to minimized additional costs combined with an overall reduced environmental impact. Aiming to develop a reproducible protocol, a premixed mortar and a selected commercially available GO were used. According to the herein reported characterization, the nanofiller consists of stacked sheets characterized by lateral size up to 600 nm × 800 nm, average thickness of 2 nm, and oxygen content of 31%. As expected, the presence of GO resulted in flowability and workability loss associated with increased rheological parameters of the fresh admixtures. However, interestingly, all hardened GO-nanocomposites showed density enhancement (up to +14% at 28 days) and improved mechanical properties (up to +29% compressive strength and +13% bending strength at 28 days). Moreover, GO-engineered samples clearly showed decreased permeability in terms of reduced water uptake and chloride ion penetration, which improves durability, compared to the reference sample. 

It can be concluded that the specific features of the selected GO allow the achievement of an adequately uniform dispersion resulting, among other effects, in the development of strong interfacial adhesion between the 2D GO flakes and the cementitious matrix, combined with the refinement of capillary pores and the development of tortuous porous paths that hamper the water ingress.

## Figures and Tables

**Figure 1 nanomaterials-11-03278-f001:**
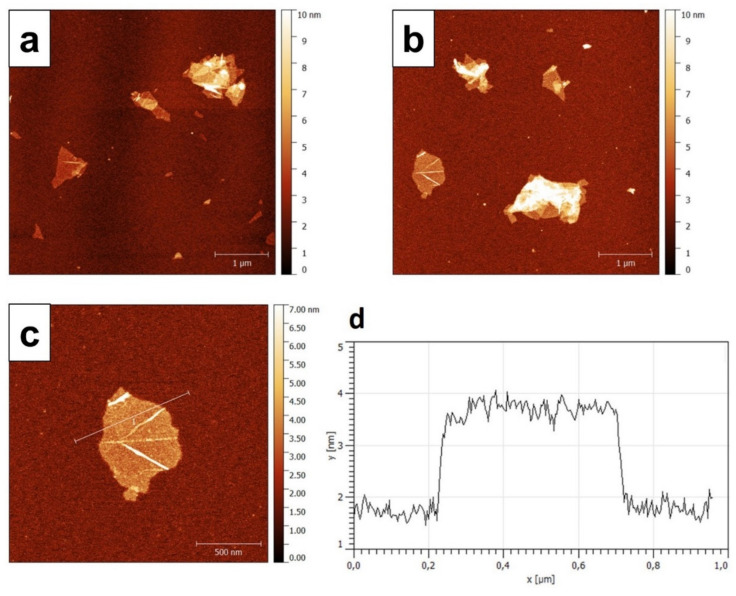
(**a**,**b**) Atomic Force Microscopy (AFM) images of Graphene Oxide (GO), (**c**,**d**) magnification and size profile of a single GO flake.

**Figure 2 nanomaterials-11-03278-f002:**
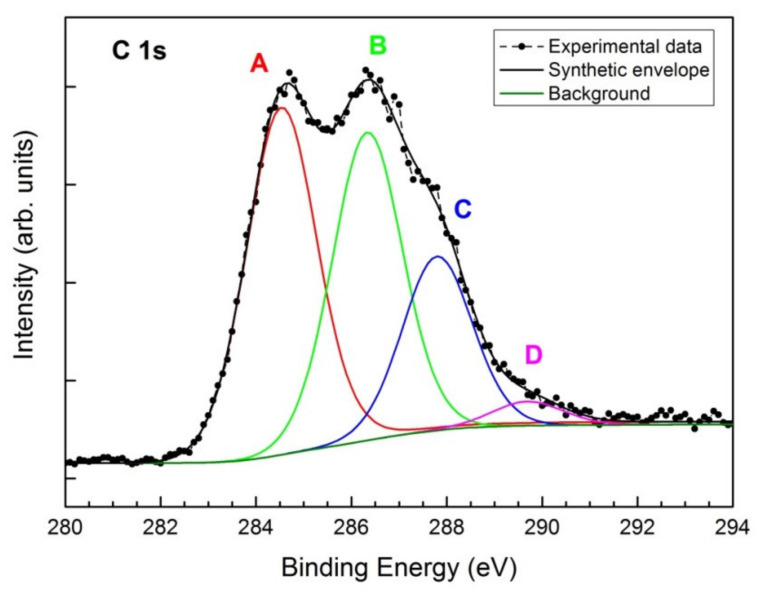
X-ray Photoelectron Spectroscopy (XPS) peak fitting of C 1s spectrum: *component A* at 284.6 eV (C-C), *component B* at 286.4 eV (C-O), *component C* at 287.9 eV (O=C−) and *component D* at 289.8 eV (O-C=O).

**Figure 3 nanomaterials-11-03278-f003:**
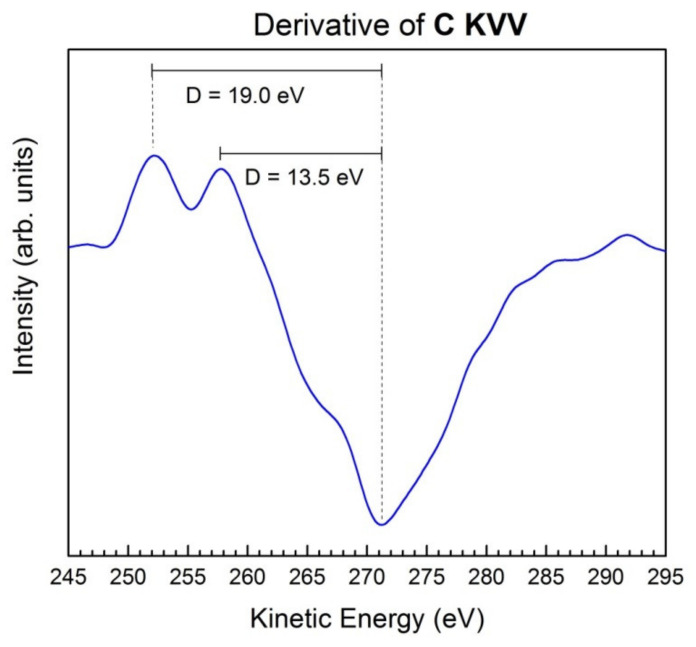
First derivative of the C KVV spectrum showing the typical D parameters of graphene (13.5 eV) and GO (19.0 eV).

**Figure 4 nanomaterials-11-03278-f004:**
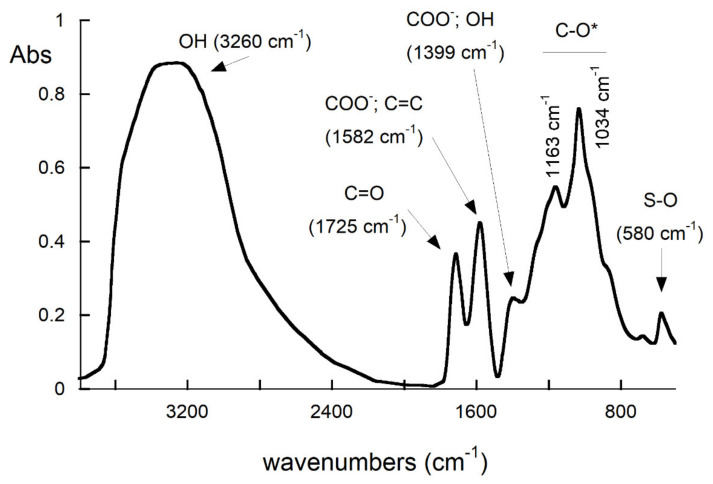
Fourier-transform infrared spectroscopy (FTIR) spectrum of GO.

**Figure 5 nanomaterials-11-03278-f005:**
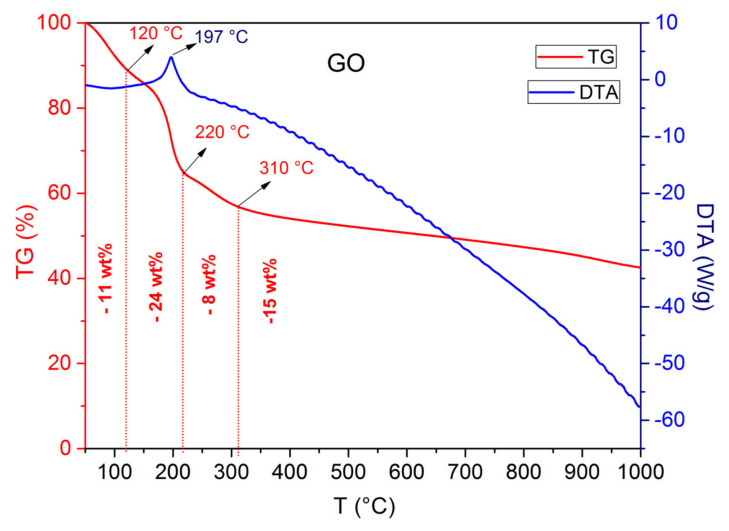
Thermogravimetric (TG) and differential thermal analysis (DTA) curves of GO.

**Figure 6 nanomaterials-11-03278-f006:**
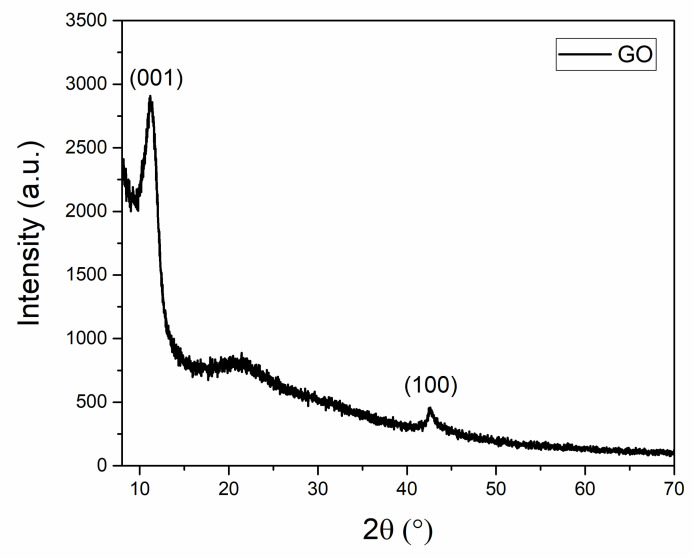
X-ray diffraction (XRD) pattern of GO.

**Figure 7 nanomaterials-11-03278-f007:**
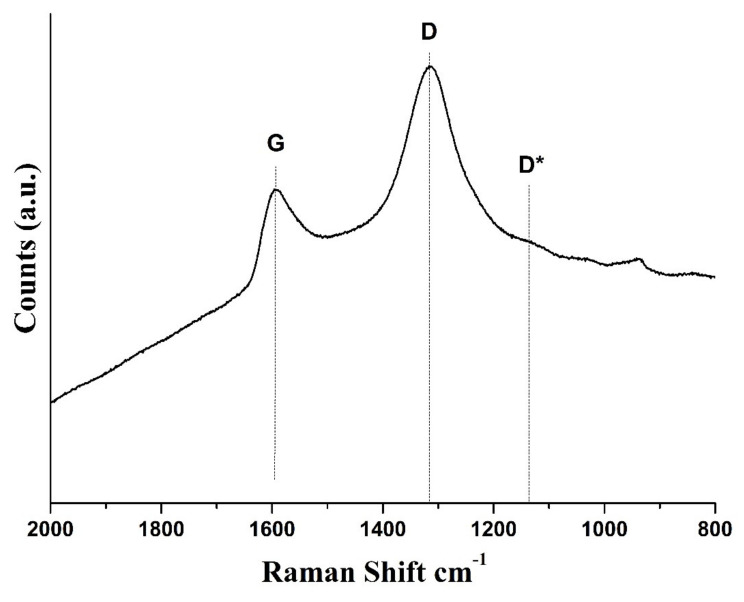
Raman spectrum of GO.

**Figure 8 nanomaterials-11-03278-f008:**
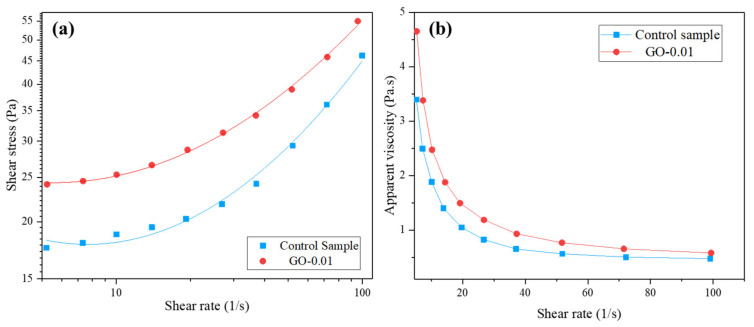
Rheological behavior of control sample and GO-modified fresh admixture (GO-0.01): (**a**) shear stress *versus* shear rate, (**b**) apparent viscosity *versus* shear rate.

**Figure 9 nanomaterials-11-03278-f009:**
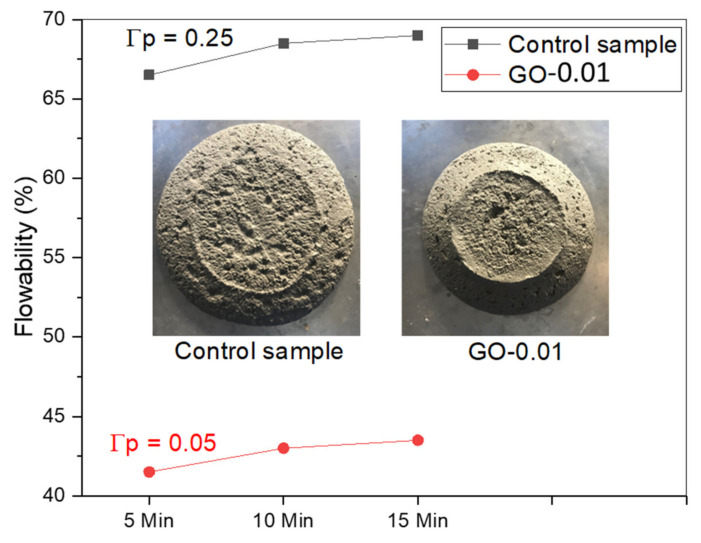
Flowability of fresh GO-modified admixture (GO-0.01) *versus* control sample.

**Figure 10 nanomaterials-11-03278-f010:**
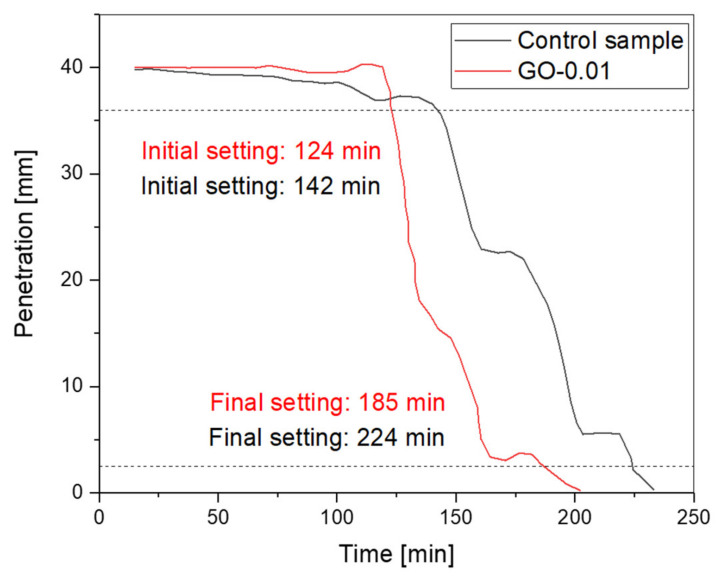
Setting time of fresh GO-modified admixture (GO-0.01) *versus* control sample.

**Figure 11 nanomaterials-11-03278-f011:**
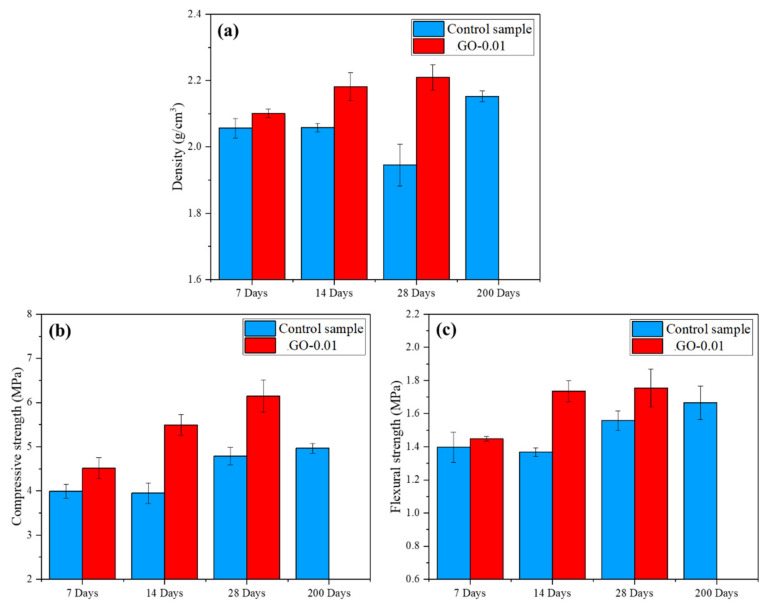
(**a**) Density, (**b**) compressive strength, (**c**) flexural strength of mortars modified with 0.01 wt.% GO and cured at 7, 14, and 28 days *versus* control sample. Properties of the control sample at 200 days are also shown.

**Figure 12 nanomaterials-11-03278-f012:**
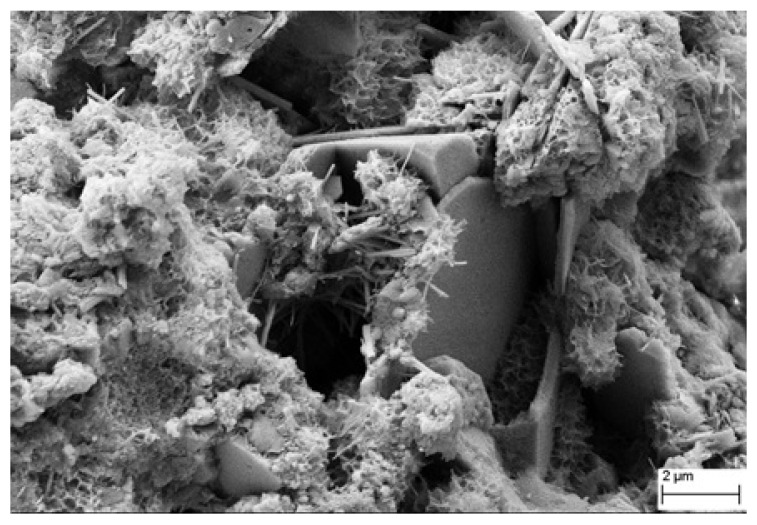
Scanning electron microscopy (SEM) micrograph of GO-modified mortar at 28 days.

**Figure 13 nanomaterials-11-03278-f013:**
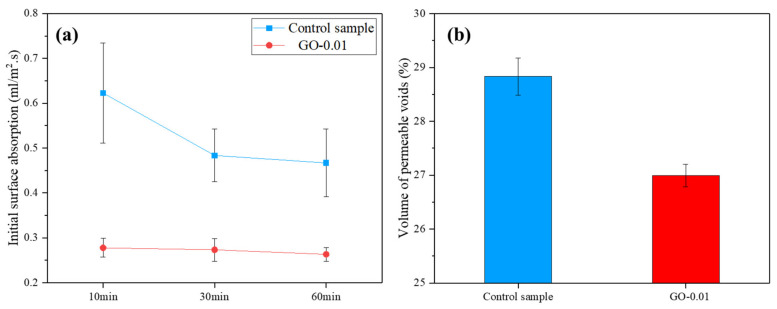
(**a**) Initial Surface Absorption Test (ISAT) and (**b**) Volume of Permeable Voids (VPV) of GO-modified mortar cured at 28 days *versus* control sample.

**Figure 14 nanomaterials-11-03278-f014:**
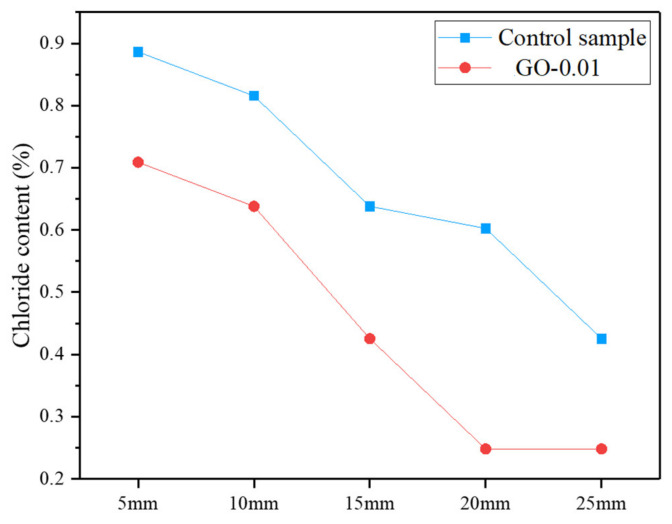
Chloride ion penetration at different depths of GO-modified mortar cured at 28 days *versus* control sample.

**Table 1 nanomaterials-11-03278-t001:** Chemical composition of GO determined by XPS.

Peak	BE (eV)	FWHM (eV)	Atomic %	Chemical Bond
C1s-A	284.6	1.7	28.4	C-C
C1s-B	286.4	1.7	24.0	C-O (epoxides, hydroxyls)
C1s-C	287.9	1.7	13.6	C=O (carboxylic acids, ketones)
C1s-D	289.8	1.7	1.9	O-C=O (carboxylates)
O1s	532.8	2.5	30.6	C-O, C=O, sulphate
S2p	168.9	2.3	1.6	sulphate

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
