# Peer review of "Extra-Low Dosage Graphene Oxide Cementitious Nanocomposites: A Nano- to Macroscale Approach"

_nanomaterials, 2021, doi:10.3390/nano11123278_

Round 1

Reviewer 1 Report

The paper is well-written, but I miss a deeper discussion of results, comparing with other similar publications. Good starting point for discussion could be the published statements on line 400. E.g. - I do not think, that - in your system - point 1) is applicable.

I don´t understand, why as the cement-lime mortar was chosen as the matrix.

The Conclusions should be more condensed; part of the current Conclusions should be used in "Discussion" - e.g. the statements about CO2 emissions.

Reviewer 2 Report

I appreciated the work devoted to the study of using of graphene oxide in the cementitious materials described in the paper. However, some important deficiencies have been observed; below are some of them presented:

  1. The obtained materials are not nanocomposites because the content of GO is very low.
  2. The sentence “In order to stop the hydration, all samples were oven-dried for 72 hours at 110 °C and stored in ethanol.” is wrong. Correctly is “In order to stop the hydration, all samples were immersed in ethanol (you will precise the period of time) and then oven-dried for 72 hours at 110 °C.”.
  3. Because both materials used in the study are commercial, I do not understand why a very broad description is made for GO, while very little is said about the M5-mortar; the mineralogical composition of the binder would be interesting, for example.
  4. What was the setting time of control and GO-0.01 samples?
  5. How was determine the density of hardened mortars?
  6. How explain the authors the decrease of density for control sample from 14 days to 28 days (see fig. 10a)?
  7. What are the properties (the density, mechanical strength) at 200 days of hardening for control sample?
  8. How was appreciated the capillary porosity?
  9. The properties are influenced by including air at mixing of components of mortar. Was considerate this in appreciate of properties?
  10. The authors show the presence of GO aggregates by SEM analysis (fig. 11). Is this true? This plackets agglomeration can be e.g attributed of AFm phase, considering that the sample was dried for 72 hours at 110°C. The authors must give the EDX spectrum for that section from SEM.

Round 2

Reviewer 2 Report

What are the properties (the density, mechanical strength) at 200 days of hardening for GO-0.01 sample?
